# Physiology and Proteomic Basis of Lung Adaptation to High-Altitude Hypoxia in Tibetan Sheep

**DOI:** 10.3390/ani12162134

**Published:** 2022-08-19

**Authors:** Pengfei Zhao, Shaobin Li, Zhaohua He, Fangfang Zhao, Jiqing Wang, Xiu Liu, Mingna Li, Jiang Hu, Zhidong Zhao, Yuzhu Luo

**Affiliations:** Gansu Key Laboratory of Herbivorous Animal Biotechnology, Faculty of Animal Science and Technology, Gansu Agricultural University, Lanzhou 730070, China

**Keywords:** Tibetan sheep, lung, proteomic, high-altitude hypoxia, adaptation

## Abstract

**Simple Summary:**

As an indigenous animal living in the Tibetan plateau, the Tibetan sheep is well adapted to high-altitude hypoxia, and the lungs play an important role in overcoming the hypoxic environment. To reveal the physiological and proteomic basis of Tibetan sheep lungs during their adaptation to hypoxia, we studied the lungs of Tibetan sheep at different altitudes using light and electron microscopy and proteome sequencing. The results showed that in the lungs of Tibetan sheep occurred a series of physiological changes with increasing altitude, and some important proteins and pathways identified by proteome sequencing further support these physiology findings. These changes at the physiological and molecular levels may facilitate the adaptation of Tibetan sheep to high-altitude hypoxia. In conclusion, these findings may provide a reference for the prevention of altitude sickness in humans.

**Abstract:**

The Tibetan sheep is an indigenous animal of the Tibetan plateau, and after a long period of adaptation have adapted to high-altitude hypoxia. Many physiological changes occur in Tibetan sheep as they adapt to high-altitude hypoxia, especially in the lungs. To reveal the physiological changes and their molecular mechanisms in the lungs of Tibetan sheep during adaptation to high altitudes, we selected Tibetan sheep from three altitudes (2500 m, 3500 m, and 4500 m) and measured blood-gas indicators, observed lung structures, and compared lung proteome changes. The results showed that the Tibetan sheep increased their O_2_-carrying capacity by increasing the hemoglobin (Hb) concentration and Hematocrit (Hct) at an altitude of 3500 m. While at altitude of 4500 m, Tibetan sheep decreased their Hb concentration and Hct to avoid pulmonary hypertension and increased the efficiency of air-blood exchange and O_2_ transfer by increasing the surface area of gas exchange and half-saturation oxygen partial pressure. Besides these, some important proteins and pathways related to gas transport, oxidative stress, and angiogenesis identified by proteome sequencing further support these physiology findings, including HBB, PRDX2, GPX1, GSTA1, COL14A1, and LTBP4, etc. In conclusion, the lungs of Tibetan sheep are adapted to different altitudes by different strategies; these findings are valuable for understanding the basis of hypoxic adaptation in Tibetan sheep.

## 1. Introduction

The Tibetan plateau, with an average altitude of 4000 m above sea level, is the highest plateau on Earth with a harsh climate that is especially hypoxic, making it a perfect natural laboratory for studying animal adaptation to high-altitude hypoxia. Under such chronic hypoxic environments, many tissues and organs of indigenous animals on the Tibetan plateau undergo adaptive changes.

In comparison to low-altitude counterparts, yaks have a larger alveolar surface area, a thinner alveolar septum and an increased plasma volume with altitude [1,2]; Guinea pigs, dogs, and Andean geese at high altitudes have a higher number of alveoli, which increases the area for blood-gas exchange [3,4,5]; Tibetan sheep have a significantly higher resting respiratory rate [6]; and Tibetan people exhibited larger chest circumferences and vital capacity [7,8], blunted hypoxic pulmonary vasoconstriction responses [9], and lower hemoglobin (Hb) concentrations [10] compared to Han Chinese people. Similarly, Hb concentrations in Tibetan horses were lower than in horse breeds living at low altitude [11], but in Tibetan mastiffs and Tibetan sheep these were higher than in dog and sheep breeds at low altitude [12,13], while in pigs they increased with altitude but decreased slightly when the altitude exceeded 3600 m [14]. The changes in these indigenous animals on the Tibetan plateau as they adapt to high-altitude hypoxia are mainly related to oxygen transfer and utilization. The lungs are important organs in the process of oxygen transfer and utilization and play a key role in the adaptation to high-altitude hypoxic environments. To date, many studies on high-altitude hypoxia adaptation have either had only two altitude controls or introduced different breeds when multiple altitudes were involved. The former cannot explain whether the changes in the organism under hypoxia are linearly related to altitude; and the latter cannot determine whether the changes are caused by altitude or breed.

Tibetan sheep are one of the main economic sources for local agro-pastoralists, and they entered the Tibetan Plateau from northern China 3100 years ago and settled permanently [15], thus Tibetan sheep have become an ideal model for studying high-altitude hypoxia adaptation. To reveal whether the changes of Tibetan sheep lungs under hypoxic conditions is linearly correlated with altitude, we selected Tibetan sheep from three different altitudes (low = 2500 m, middle = 3500 m, and high = 4500 m) and measured their blood-gas indicators, observed their lung phenotypic structures using optical and electron microscopes, compared proteome changes in their lungs using a data-independent acquisition (DIA) quantitative technique, verified proteome results using a western blot analysis (WB), and studied the expression locations of key proteins using an immunofluorescence technique. By leveraging phenotypic and proteomic data of Tibetan sheep lungs from multiple altitudes, this study found that the lung structures were different and a set of proteins and molecular pathways may explain, to some extent, the physiological and proteomic basis of adaptation to high-altitude hypoxia in Tibetan sheep.

## 2. Materials and Methods

### 2.1. Ethics Statement

The animal study was approved by the Animal Care Committee at Gansu Agricultural University (approval number GAU-LC-2020-27). All experiments on these sheep were conducted according to animal protection and use guidelines established by the Ministry of Science and Technology of the People’s Republic of China (Approval number 2006-398).

### 2.2. Sample Collection and Blood-Gas Indicators Measure

In this study, six healthy female Tibetan sheep of approximately 3 years old were selected at each altitude of 2500 m (L), 3500 m (M) and 4500 m (H), respectively (Figure 1). The right lung parenchyma were rapidly collected after slaughter and one part was frozen in liquid nitrogen for proteome sequencing and a WB test, and another part was fixed in 3% glutaraldehyde for transmission electron microscopy (TEM) observation, and another part was fixed in 4% paraformaldehyde for hematoxylin-eosin (HE) and Weigert resorcinol magenta (WRM) staining, and immunofluorescence testing. The left lung was then collected to make cast specimen of the pulmonary artery for scanning electron microscopy (SEM) observation.

Including these slaughtered sheep, jugular venous blood was collected from 139, 15, and 24 Tibetan sheep at L, M, and H altitudes, respectively, using 5 mL sodium heparin tubes to acquire blood-gas indicator measurements using an i-STAT blood-gas analyzer (Abbott, Chicago, IL, USA). Blood-gas indicators include: Hb concentration, hematocrit (Hct), partial pressure of oxygen (pO_2_), and pH and oxygen saturation (sO_2_). The latter three were used according to the formula of Lichtman et al. [16] to calculate the half-saturation oxygen partial pressure (p50), which represented the pO_2_ at 50% saturation of Hb and marks the hemoglobin-oxygen affinity.

### 2.3. Observation of Lung Structure

The observation of lung structures included the number and area of alveoli identified by HE staining, the content and distribution of elastic fibers in the lungs identified by WRM staining, the ultrastructure of alveolar septum identified by TEM, and the abundance and diameter of arterioles in the lungs identified by SEM.

Three 5 mm × 5 mm × 1 mm lung parenchymal tissues fixed in 4% paraformaldehyde solution were selected from each altitude for HE staining. Samples were dehydrated in sequential 75%, 85%, 95%, and 100% ethanol, then rinsed with xylene and embedded in paraffin, the paraffin block was cut into 5µm thick sections using a Leica-2016 rotary microtome (Leica, Wetzlar, Germany), followed by HE staining. Three different 200 × micrographs fields of view of HE-stained sections were taken using a BA210Digital microscope camera (Motic, Xiamen, China) for alveoli counting and area measurements using Image-Pro Plus 6.0.

One paraffin section was selected from each altitude for WRM staining, and three different 400 × micrographs fields of view of WRM-stained sections were taken using a BA210Digital microscope camera (Motic, Xiamen, China) for the observation of content and distribution of elastic fibers. Both HE and WRM staining are performed by Lilai Biotech Co., Ltd. (Chengdu, China).

One 2 mm × 2 mm × 1 mm section of lung parenchymal tissue fixed in 3% glutaraldehyde solution at 4 °C was selected from each altitude for TME observation. Samples were refixed with 1% osmium tetroxide and dehydrated in sequential 30%, 50%, 70%, 80%, 90%, 95%, and 100% acetone, then soaked with a mixture of 3:1, 1:1, and 1:3 acetone and epoxy successively, for 40 min per step, and embedded in an epoxy resin. Then they were cut into ultrathin sections using an EM-UC7 microtome (Leica, Wetzlar, Germany) and stained with uranyl acetate and lead citrate, and the ultrastructure of the stained sections of the alveolar septum was observed using a JEM-1400PLUS TEM (Jeol, Tokyo, Japan).

The left lungs of six sheep from each altitude were selected to make cast specimens of the pulmonary artery. Ten percent and fifteen percent casting agents were prepared by dissolving acrylonitrile-butadiene-styrene pellets in 1:1 acetone and butanone, which were injected from the aorta of the left lungs successively, until the arteries were full. We waited for the casting agent to solidify and then placed them in 30% hydrochloric acid to corrode. The best cast specimens from each altitude were selected and sprayed with gold using an E-1045 ion coater (Hitachi, Tokyo, Japan), and the abundance and diameter of arterioles were observed using an S-3400N SEM (Hitachi, Tokyo, Japan).

### 2.4. Sample Preparation for Mass Spectrometry

Four Tibetan sheep were selected from each altitude and the total proteins were extracted from the parenchymal tissue of the lungs. First, ~50 mg of each sample was transferred into a lysis buffer (2% sodium dodecyl sulphate, 7 M urea, and 1 mg/mL protease inhibitor cocktail), and homogenized for 3 min, three times, at 4 °C using an ultrasonic homogenizer. The homogenate was centrifuged at 4 °C for 15 min at 15,000 rpm and the supernatant was collected.

The extracted proteins (50 μg) were suspended in 50 μL solution and reduced to 55 °C for 1 h by adding 1 μL 1 M dithiothreitol, they were alkylated in the dark at 37 °C for 1 h by adding 5 μL of 20 mM iodoacetamide, then precipitated using 300 μL cold acetone at −20 °C, overnight. The precipitate was washed two times using a prechilled acetone and resuspended in 50 mM ammonium bicarbonate. Finally, the proteins were digested with a sequence-grade modified trypsin (Promega, Madison, WI, USA) at a substrate/enzyme ratio of 50:1 (*w*/*w*) at 37 °C for 16 h.

The peptide mixtures were redissolved in buffer A (20 mM aqueous ammonium formate, adjusted to pH = 10.0 with ammonium hydroxide) and separated at a high pH using an Ultimate 3000 system (ThermoFisher, Waltham, MA, USA) connected to a reverse phase XBridge C18 column (Waters, Milford, MA, USA). Separation was performed using a linear gradient from 5% to 45% solvent B (80% acetonitrile with 20 mM ammonium formate, adjusted to pH = 10.0 with ammonium hydroxide) for 40 min.

### 2.5. Data-Dependent Acquisition (DDA) Qualitative Database Construction and Spectral Library

The peptides were redissolved in 30 μL solvent A (0.1% formic acid aqueous solution) and analyzed using an LC-MS/MS on an Orbitrap Fusion Lumos coupled to an EASY-nLC 1200 system (ThermoFisher, Waltham, MA, USA). Three-microliter samples were loaded onto the analytical column (Acclaim PepMap C18, 75 μm × 25 cm) and separated within 120 min with a gradient from 5% to 35% B (0.1% formic acid in acetonitrile). The column flow rate was controlled at 200 nL/min and kept at 40 °C and the electrospray voltage was 2 kV and the inlet of the mass spectrometer was used. The Orbitrap Lumos mass spectrometer was run under DDA mode, and automatically switched between MS and MS/MS acquisition. The mass spectrometry parameters were set as follows: (1) MS—scan range (*m*/*z*) = 350–1200; resolution = 120,000; AGC target = 400,000; max injection time = 50 ms; filter dynamic exclusion duration = 30 s; (2) HCD-MS/MS—resolution = 15,000; AGC target = 50,000; max injection time = 35 ms; and collision energy = 32.

Raw data of the DDA were analyzed using Spectronaut X (Biognosys AG, Zurich, Switzerland) with its default settings to generate an initial target list. The software was set up to search the ovine database and contaminant database and assumed trypsin was the digestive enzyme based on the ovine genome assembly, Oar_rambouillet_v1.0 (accessed on 2 November 2017, https://www.ncbi.nlm.nih.gov/assembly/GCF_002742125.1). Search parameters were as follows: fixed modification—Carbamidomethyl (C); and variable modification—oxidation (M). The *q*-value (*p*-value corrected by the false discovery rate (FDR) [17]) cutoff on the precursor and protein levels was applied at 1%.

### 2.6. DIA Data Collection and Analysis

The Orbitrap Lumos mass spectrometer was run under DIA mode and automatically switched between MS and MS/MS modes for DIA data collection. The parameters were: (1) MS—scan range (*m*/*z*) = 350–1200; resolution = 120,000; AGC target = 1,000,000; maximum injection time = 50 ms; (2) HCD-MS/MS—resolution = 30,000; AGC target = 1,000,000; collision energy = 32; stepped CE = 5%; (3) DIA was performed with a variable isolation window, where each window overlapped 1 *m*/*z*, and the window number was 60.

Raw data of DIA were analyzed using Spectronaut X (Biognosys AG, Zurich, Switzerland) with its default settings. The *q*-value cutoff on the precursor and protein levels was at 1%. Decoy generation was set as mutated. All selected precursors passing the filters were used for quantification, and the average top three filtered peptides were used to calculate the major group quantities. After Student’s *t*-tests, differentially expressed proteins (DEPs) were filtered between different altitudes if *q* < 0.05 and fold change ≥1.5 or ≤0.66.

The DEPs were annotated and classified by Gene Ontology (http://geneontology.org/) (GO) and Kyoto Encyclopedia of Genes and Genomes (http://www.genome.jp/kegg/). (KEGG) pathway functional annotations including biological process (BP), cellular component (CC), and molecular function (MF). The significant enriched GO terms and KEGG pathways were defined (*p* < 0.05) using Fisher’ exact [18].

To characterize the expression patterns of DEPs, a cluster analysis was performed using Short Time-series Expression Miner (STEM) software [19], and the clustered profiles with *p* < 0.05 were considered as significant profiles. The DEPs were used to conduct protein-protein interaction (PPI) network analysis using the STRING database (http://string-db.org/) [20], where the confidence score set to high (>0.7), and the interaction network was constructed using Cytoscape software [21].

### 2.7. WB and Immunofluorescence

To detect the expression level of the key protein (HBB) in lung parenchyma tissues, proteins were extracted from three Tibetan sheep at each altitude and separated on 12% SDS-PAGE gel. They were then transferred to polyvinylidene fluoride (PVDF) membranes, and these membranes were blocked and probed against HBB with rabbit polyclonal antibody (Abmart, Shanghai, China) at 4 °C overnight. Anti-β-tubulin polyclonal antibody (Abmart, Shanghai, China) was used as an internal reference. These membranes were further incubated with goat anti-rabbit IgG antibody (Abmart, Shanghai, China) for 2 h at room temperature. Protein bands were visualized using NcmECL Ultra reagents (NCM Biotech, Suzhou, China) in an X-ray room, and quantified using the AlphaEase FC software.

To detect the expression locations of HBB in lung parenchyma tissues, one paraffin section was selected from each altitude for immunofluorescence staining. Sections were blocked with 3% BSA (Servicebio, Wuhan, China) and incubated with rabbit polyclonal antibody against HBB (Abmart, Shanghai, China) at 4 °C overnight. Then, sections were incubated with goat anti-rabbit IgG antibody labeled with FITC (Abmart, Shanghai, China) for 50 min at room temperature and counterstained with DAPI. Finally, these sections were seen using an Eclipse C1 fluorescence microscope (Nikon, Tokyo, Japan) and imaged using CaseViewer software.

### 2.8. Statistical Analyses

Statistical analyses were performed using SPSS 19.0. The significance of the differences between the blood-gas indicators, proportion to the alveoli in the field of view area, number and average area of alveoli, and WB quantification results between different altitudes were determined using a one-way analysis of variance (ANOVA) followed by Duncan’s test. The results are expressed as mean ± SD, where the statistical significance was set at *p* < 0.05.

## 3. Results

### 3.1. Differences in Blood-Gas Indicators

The measurements of the blood-gas indicators show that the Hb concentration and Hct first increased (*p* < 0.05) and then decreased (*p* < 0.05) with increasing altitude; while p50 had the opposite trend, pO_2_ and sO_2_ showed a decreasing trend (*p* < 0.05) with increasing altitude (Figure 2).

### 3.2. Structural Differences in the Lungs

The HE staining results showed that the proportion of alveoli to the field of view area gradually increased with increasing altitude, but the difference was not significant (*p* > 0.05) (Figure 3A). The number and average area of alveoli were higher (*p* < 0.05) and lower (*p* < 0.05) at H altitudes than at M and L altitudes (Figure 3B,C).

WRM can dye elastic fibers to dark blue, and the results showed that the size of the elastic fibers increased with altitude (Figure 4).

The cast specimens of the pulmonary arteries were observed using SEM, and the results showed that the branches of the pulmonary arteries increased and became thinner in diameter with increasing altitude (Figure 5).

Observation of the alveolar septum by TEM revealed a thinning of the air-blood barrier with increasing altitude (Figure 6).

### 3.3. Summary of the Proteomics Data

Using the spectral library constructed using raw data from the DDA as a reference, precursors (65,443), peptides (53,319), protein groups (5093), and proteins (7572) were identified (using *q* < 0.01 as precursor/protein threshold) (Figure 7A), and a total of 7487, 7491, and 7504 proteins were identified in lungs of Tibetan sheep from L, M, and H altitude, respectively, with 7378 proteins being expressed at all three altitudes (Figure 7D). More than two-thirds of the proteins consisted of ten or fewer peptides, and approximately one-third of the proteins consisted of more than eleven or more peptides (Figure 7B). After protein quantification normalization, the strength of the signals in twelve samples showed similar response intensities (Figure 7C).

A total of 149, 71, and 137 DEPs were identified when comparing L and M, M and H, and L and H altitudes (*q* < 0.05 and fold change ≥ 1.5 or ≤ 0.66). Of these, 66, 22, and 49 proteins were significantly up-regulated and 83, 49, and 88 proteins were significantly down-regulated, respectively (Figure 8).

### 3.4. GO Enrichment Analysis of the DEPs

In order to understand the functions of DEPs, a GO enrichment analysis was performed, and it was found that the 149 DEPs between L and M altitudes were significantly enriched in 357 BP terms, 49 CC terms, and 49 MF terms. The top five significant results with the lowest *p*-values were: response to bacterium (*p* < 0.001), extracellular region (*p* < 0.001), multicellular organismal process (*p* < 0.001), extracellular region part (*p* < 0.001) and response to other organisms (*p* < 0.001). Besides these, important GO terms such as blood circulation (*p* < 0.001) and gas transport (*p* < 0.01) were also identified (Figure 9A).

Concerning the 71 DEPs between M and H, they were significantly enriched in 228 BP terms, 28 CC terms, and 42 MF terms. The top five significant results with the lowest *p*-values were: extracellular region part (*p* < 0.001), extracellular region (*p* < 0.001), FACIT collagen trimer (*p* < 0.001), extracellular matrix (*p* < 0.001), and vesicle (*p* < 0.001). In addition to these, important GO terms such as cellular response to oxidative stress (*p* < 0.001) and collagen metabolic process (*p* < 0.001) were also identified (Figure 9B).

Concerning the 137 DEPs between L and H, they were significantly enriched in 433 BP terms, 57 CC terms, and 42 MF terms. The top five significant results with the lowest *p*-values were: extracellular region part (*p* < 0.001), extracellular region (*p* < 0.001), FACIT collagen trimer (*p* < 0.001), extracellular structure organization (*p* < 0.001), and extracellular matrix organization (*p* < 0.001). In addition to these, important GO terms such as reactive oxygen species (ROS) metabolic process (*p* < 0.001) and cardiovascular system development (*p* < 0.001) were also identified (Figure 9C). Notably, the number of significantly enriched terms is higher than the number of DEPs. This may be due to the same protein having multiple functions, so it is enriched into multiple terms.

### 3.5. KEGG Pathway Analysis of the DEPs

The KEGG pathway analysis of the 149 DEPs between L and M altitudes showed that drug metabolism, cytochrome P450 (*p* < 0.001), retinol metabolism (*p* < 0.001), and metabolism of xenobiotics by cytochrome P450 (*p* < 0.001) were top three enriched pathways. Besides these, important pathways such as hematopoietic cell lineage (*p* < 0.05) and ECM-receptor interaction (*p* < 0.05) were also identified. Concerning the 71 DEPs between M and H altitudes, the top three enriched pathways were drug metabolism, other enzymes (*p* < 0.001), pantothenate, and CoA biosynthesis (*p* < 0.001), and protein digestion and absorption (*p* < 0.001). Besides these, important pathways such as glutathione metabolism (*p* < 0.001) was also identified. Concerning the 137 DEPs between L and H altitudes, the top three enriched pathways were drug metabolism, other enzymes (*p* < 0.001), nicotinate and nicotinamide metabolism (*p* < 0.001), and pyrimidine metabolism (*p* < 0.001). Besides these, important pathways such as hematopoietic cell lineage (*p* < 0.05) and glutathione metabolism (*p* < 0.05) were also identified (Figure 10).

### 3.6. Cluster Analysis of the DEPs

A cluster analysis was performed to characterize the expression patterns of DEPs, and eight expression patterns were yielded. Two of which were significantly enriched with increasing altitude (*p* < 0.05); one (profile 1) showed a significant down-regulation of DEPs with increasing altitude, while the other (profile 6) showed a significant up-regulation of DEPs with increasing altitude (Figure 11).

### 3.7. Interactive Network Analysis of the DEPs

A PPI analysis was performed to explore the PPI networks altered in the lungs of Tibetan sheep with increasing altitude, and the interaction score required a minimum of greater than 0.7. Fifty-three of one hundred and forty-nine DEPs between L and M altitudes constituted an interaction network. This network contained important proteins related to the oxidative stress response (for example, GPX3, ALOX5AP, ALOX15, and MPO), gas transport (for example, HBB, HP, and AHSP), and vasculature development (for example, ITGA5, FLNC, FLNA, FN1, COL6A1, COL6A2, and COL6A5) (Figure 12A). Of the 71 DEPs between M and H altitudes, four constituted an interaction network related to the oxidative stress response (GSTA1, GPX3, TXNRD1, and ADH1C) (Figure 12B). Of the 137 DEPs between L and H altitudes, 37 constituted an interaction network related to the oxidative stress response (for example, GSTA1, GPX3, ALOX15, MPO, and CRP) and vasculature development (for example, ITGA5, FLNC, FLNA, and COL6A5) (Figure 12C).

### 3.8. WB and Immunofluorescence Validation of DIA Data

To confirm the DIA results, we performed a WB and an immunofluorescence analysis for HBB protein, the protein β-tubulin was used as an internal reference. The expression levels of HBB increased from L to M altitudes (*p* < 0.05), while they decreased from M to H altitudes (*p* < 0.05) (Figure 13). The results of a WB and an immunofluorescence analysis of the HBB protein were consistent with our DIA data.

## 4. Discussion

Adaptation to high-altitude hypoxia in Tibetan sheep is accompanied by changes in blood-gas indicators. In this study, measurements of the blood-gas indicators in 139, 15, and 24 Tibetan sheep at L, M, and H altitudes showed that the pO_2_ and sO_2_ levels decreased with increasing altitude (*p* < 0.05), and p50 first decreased (*p* < 0.05) and then increased (*p* < 0.05) with increasing altitude, while Hb concentration and Hct had the opposite trend (Figure 2). When the atmospheric pressure decreased with altitude, the ambient pO_2_ also decreased, causing low-pressure hypoxia in Tibetan sheep, namely, lower levels of pO_2_ in their blood, and consequently a decrease in sO_2_ levels [22]. The p50 indicates hemoglobin-oxygen affinity; a high p50 means that O_2_ is more readily released into the tissues by dissociation with hemoglobin, and the decreased pO_2_ and sO_2_ levels facilitate this process. This compensates for the effects of lower Hb concentrations and Hct on Tibetan sheep at a H altitude. Consistent with this study, the Hb concentrations in pigs first increased with altitude and then decreased when the altitude was over 3600 m [14]. Similarly, the Hb concentrations in Tibetan people were lower than in lowlanders [10], and in the Tibetan horse, both Hb concentrations and Hct were lower than in horse breeds living at a low altitude [11]. Appropriate increases in Hb concentration and Hct can enhance one’s O_2_-carrying capacity, for example, Hb concentrations were higher in dogs and pigs at 3000 m altitude than in their lower altitude counterparts [12,14]. However, excessive Hb concentrations and Hct increases blood viscosity and may lead to pulmonary hypertension. Hence, at higher altitudes animals may overcome hypoxia by means other than increasing Hb concentrations and Hct, for example, increasing pulmonary ventilation [23], blood volume [24], heart rate [25], and p50. An increased heart rate requires lower Hb concentration and Hct to reduce blood viscosity to avoid pulmonary hypertension, therefore, Hb concentration and Hct tended to be lower in animals at high altitudes than in animals at middle and even low altitudes. As might be expected, in this study, the Hb concentration and Hct first increased and then decreased with increasing altitude in Tibetan sheep.

In addition to the blood-gas indicators, the physiological structure of the lungs of Tibetan sheep also changed with altitude as they are adapted to high-altitude hypoxia, including changes to the alveoli, blood vessels, and the air-blood barrier. In this study, the proportion of alveoli to the field of view area gradually increased (*p* > 0.05), the number and average area of alveoli increased (*p* < 0.05) and decreased (*p* < 0.05) with increasing altitude (Figure 3), resulting in an increase in the total surface area for gas exchange with increasing altitude. Consistent with this study, guinea pigs [3] and dogs [4] at high altitudes had higher alveolar counts, whereas Andean geese [5] and yaks [2] at high altitude, had larger alveolar surface areas. This suggests that the increased surface area for gas exchange in animals at high altitudes is a compensatory response to the limited availability of O_2_ in high-altitude environments. Besides the changes in alveoli with increasing altitude, the elastic fibers and abundance of blood vessels increased in the lungs of Tibetan sheep (Figure 4 and Figure 5). High-altitude hypoxic environments leads to increased plasma volume and heart rate [24,25], and consequently an increase in blood pressure. Mammals inhabiting high altitudes avoid pulmonary hypertension by having a good vasodilatation and vasoconstriction capacity and large vascular volumes, in other words, they promote vascular fibrosis and intense vascularization [5,26], which also occurred in this study. An increase in alveolar number and vascular abundance increased the surface area for gas exchange, and a thinner alveolar septum, which is the site of gas exchange, leads to more efficient gas exchange. In this study, the alveolar septum became thinner with increasing altitude (Figure 6), and similar results were also found in yaks [1].

To reveal the molecular basis of the physiological differences in the lungs of Tibetan sheep at three altitudes, this study used the DIA quantitative technique of comparing proteome changes of twelve lungs at three altitudes, and a total of 267 DEPs were identified. These DEPs are mainly related to gas transport, oxidative stress, and angiogenesis. The DEPs, HBB, PRDX2, AHSP, HP, SPTB, and DMTN, were related to gas transport. HBB is the subunit that makes up Hb, which binds and transports O_2_ and CO_2_ and maintains an acid-base balance. Lefrancais et al. [27] found that the lungs may have considerable hematopoietic potential, and the present study also found that HBB was in high abundance in the lungs of Tibetan sheep. It is worth noting that the high abundance of HBB in the lungs may come from both the lung’s own hematopoiesis and the red blood cells retained in the pulmonary capillaries. PRDX2 and AHSP maintain erythrocyte function by removing hydrogen peroxide from erythrocytes [28,29] and stabilizing Hb in erythrocytes [30]. Hb is extremely toxic when released from erythrocytes, and haptoglobin (HP) can bind and clear free Hb to protect cells from oxidative damage [31]. The expression patterns of these four proteins were clustered into profile 5 (Figure 11), consistent with the trend of Hb concentration and Hct with increasing altitude; that is, they increased from L to M altitudes, and decreased from M to H altitudes. This suggests that the changes in the blood-gas indicators could be explained to some extent by the changes in the abundance of these proteins. SPTB and DMTN are necessary for erythrocyte membrane stability [32,33] and were clustered in profile 6 (Figure 11), that is, the abundance increased at H altitudes. Together, these proteins are involved in gas transport and maintain the stability of erythrocytes.

The DEPs, GPX1, GPX3, GSTA1, and GSTA2, were related to oxidative stress. GPX1 and GPX3 are intracellular antioxidant enzymes that reduce ROS, such as hydrogen peroxide, to water to minimize their harmful effects [34,35]. GSTA1 and GSTA2 are widespread enzymes that play a central role in the counteraction of oxidative stress and detoxification, particularly the scavenging of hydrogen peroxide [36,37]. The expression patterns of these proteins were all clustered into profile 3 (Figure 11), that is, their abundance is reduced at H altitudes which may lead to ROS accumulation. Previous research supports this speculation: ROS production was increased with altitude [38,39]. High concentrations of ROS can cause endothelial cell damage, but appropriate concentrations of ROS, such as hydrogen peroxide, generally promote angiogenesis [40,41,42]. It is therefore hypothesized that one of the reasons for the higher abundance of lung vessels in Tibetan sheep at H altitudes is due to the reduced abundance of these four proteins (Figure 5).

In addition to this, DEPs FLII, HRG, TAGLN, MPO, GSN, GNG2, CA2, COL14A1, LTBP4, MFAP5, and EHD4 were also related to angiogenesis and vascular function. Among these proteins, FLII, HRG, and TAGLN have anti-angiogenic effects, the attenuation of FLII promotes angiogenesis and wound healing [43,44], angiogenesis is enhanced in HRG-deficient mice [45], and the genetic disruption of TAGLN enhanced angiogenesis [46]. TAGLN also facilitates angiotensin II-induced vascular contraction [47], similar to this, MPO is an enzyme abundantly expressed in neutrophils and has potent vasoconstrictive properties [48]. The deficiency of GSN results in an increased pulmonary vascular permeability [49], and consequently an increase in the air-blood exchange efficiency. The expression patterns of these five proteins that were just mentioned were clustered into profile 0 or 1 (Figure 11), meaning the abundance of these proteins reduced at H altitudes and this may lead to promoted angiogenesis, avoidance of pulmonary hypertension, and increased vascular permeability in the lungs of Tibetan sheep at H altitudes.

Decreased GSN levels increases vascular permeability, whereas GNG2 is directly involved in increases in vascular permeability [50], and also maintaining arterial diastole [51]. CA2 can reduce nitrite to NO [52], similar to the way GNG2 acts as a vasodilator. COL14A1 and LTBP4 are the basic components of the extracellular matrix and promote the elastic fiber network and angiogenesis [53,54,55]. MFAP5, by blocking notch signaling in endothelial cells, initiates angiogenesis [56], and EHD4 is recruited by PACSIN2 to participate in angiogenesis [57]. The expression patterns of these six proteins were clustered into profiles 4, 6, or 7 (Figure 11), that is, they have a higher abundance at H altitudes. It was therefore hypothesized that the increased angiogenesis and elastic fibers in the lungs of Tibetan sheep at H altitudes was due to the increased expression of COL14A1, LTBP4, MFAP5, and EHD4 and the decreased expression of FLII, HRG, and TAGLN. The avoidance of pulmonary hypertension is due to the increased expression of GNG2 and CA2 and the decreased expression of TAGLN and MPO, and the increased vascular permeability is due to the increased expression of GNG2 and the decreased expression of GSN.

GO functional enrichment and a KEGG pathway analysis of the DEPs between the three altitudes revealed that most of the enriched terms and pathways were also associated with gas transport, oxidative stress, and angiogenesis (Figure 9 and Figure 10). These GO terms were blood circulation (*p* < 0.001) and gas transport (*p* < 0.01), cellular response to oxidative stress (*p* < 0.001), and ROS metabolic process (*p* < 0.001), collagen metabolic process (*p* < 0.001), and cardiovascular system development (*p* < 0.001). These KEGG pathways were hematopoietic cell lineage (*p* < 0.05), glutathione metabolism (*p* < 0.001), protein digestion and absorption (*p* < 0.001), and ECM-receptor interaction (*p* < 0.05). As might be expected, the PPI network in Figure 12 was also involved in gas transport, oxidative stress responses, and vasculature development. The results of the WB and immunofluorescence techniques were consistent with the DIA data (Figure 13), suggesting that DIA data are reliable to quantify the protein abundance in Tibetan sheep lung in this study. Together, these findings at the proteomic level explain, to some extent, the physiological changes observed in Tibetan sheep when overcoming high-altitude hypoxia.

## 5. Conclusions

In conclusion, the lungs of Tibetan sheep are adapted to different altitudes by different strategies. At M altitudes, Tibetan sheep increased their O_2_-carrying capacity by increasing the Hb concentration and Hct. At H altitudes, to avoid pulmonary hypertension, Tibetan sheep decreased the Hb concentration and Hct, while they increased the efficiency of air-blood exchange and O_2_ transfer by increasing the surface area of gas exchange and p50. Besides these, some important proteins and pathways related to gas transport, oxidative stress, and angiogenesis are identified by proteome sequencing and they further support these physiology findings.

## Figures and Tables

**Figure 1 animals-12-02134-f001:**
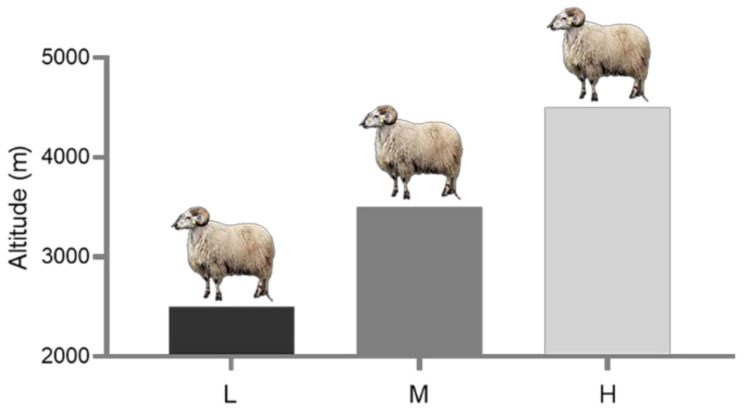
Altitude distribution of Tibetan sheep: low (L) altitude = 2500 m; middle (M) altitude = 3500 m; and high (H) altitude = 4500 m. The 2500 m groups were from Zhuoni County, Gannan Tibetan Autonomous Prefecture, China; the 3500 m groups were from Haiyan County, Haibei Tibetan Autonomous Prefecture, China; and the 4500 m groups were from Zhidoi County, Yushu Tibetan Autonomous Prefecture, China.

**Figure 2 animals-12-02134-f002:**
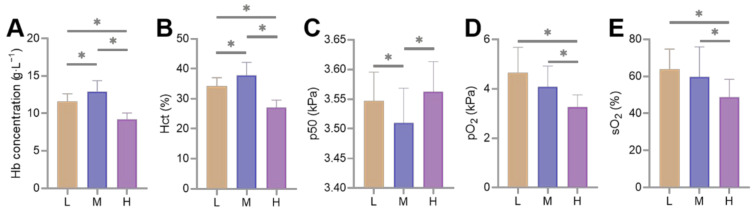
Differences in blood-gas indicators of Tibetan sheep at low (L), middle (M), and high (H) altitudes, * indicates significant differences between different altitudes (*p* < 0.05). Hemoglobin (Hb) concentration (**A**), Hematocrit (Hct) (**B**), Half-saturation oxygen partial pressure (p50) (**C**), Partial pressure of oxygen (pO_2_) (**D**) and Oxygen saturation (sO_2_) (**E**) are shown.

**Figure 3 animals-12-02134-f003:**
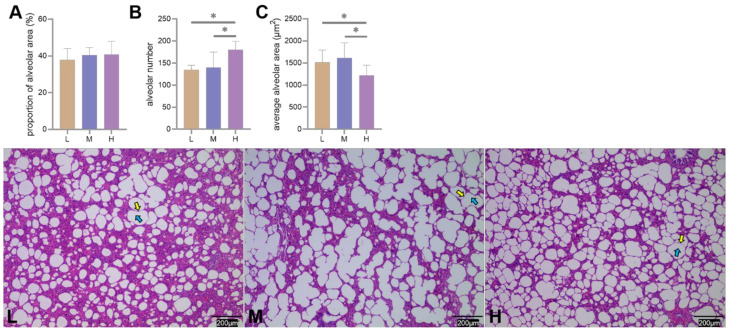
Differences in proportion of alveoli to the field of view area (**A**), number (**B**), and average area (**C**) of alveoli under 200 × field of view among low (L), middle (M), and High (H) altitudes, * indicates significant differences between different altitudes (*p* < 0.05). Alveoli under an optical microscope are shown (L–H), yellow and blue arrows indicate alveolar septum and alveoli, respectively.

**Figure 4 animals-12-02134-f004:**
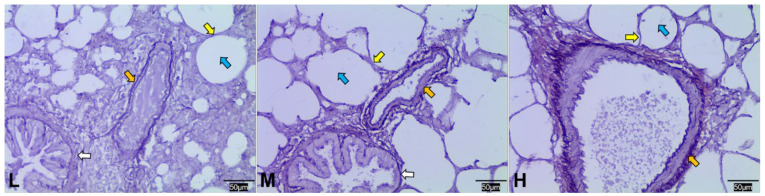
Weigert resorcinol magenta (WRM) staining of elastic fibers at low (L), middle (M), and high (H) altitudes under 400 × field of view. Yellow, blue, orange, and white arrows indicate alveolar septum, alveoli, blood vessels, and terminal bronchioles, respectively.

**Figure 5 animals-12-02134-f005:**
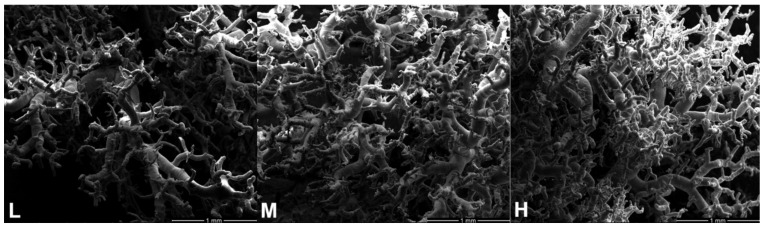
Branching characteristics of pulmonary arteries at low (L), middle (M), and high (H) altitudes under 100 × field of view.

**Figure 6 animals-12-02134-f006:**
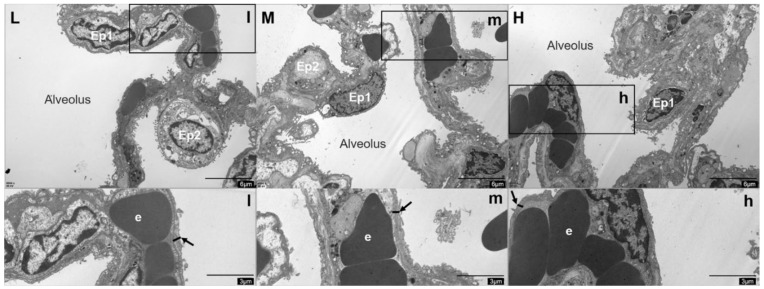
Alveolar septum ultrastructure at low (L), middle (M), and high (H) altitudes under 6000 × field of view. Type I epithelial cell (Ep1), type II epithelial cell (Ep2), and erythrocyte (e) are shown, arrows indicate air-blood barriers.

**Figure 7 animals-12-02134-f007:**
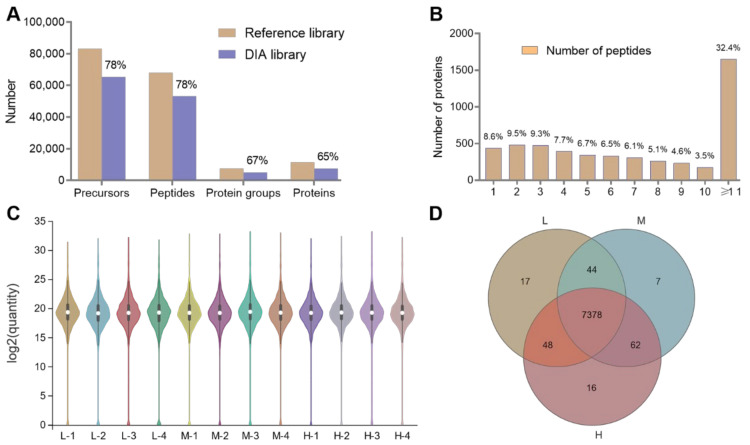
Summary of the proteomics data. Percentage of the DIA library compared to the reference library (**A**). Percentage of the number of peptides constituting the proteins (**B**). Normalization of protein quantification for samples of low (L), middle (M), and high (H) altitudes (**C**). Venn diagrams of identified proteins in low (L), middle (M), and high (H) altitudes (**D**).

**Figure 8 animals-12-02134-f008:**
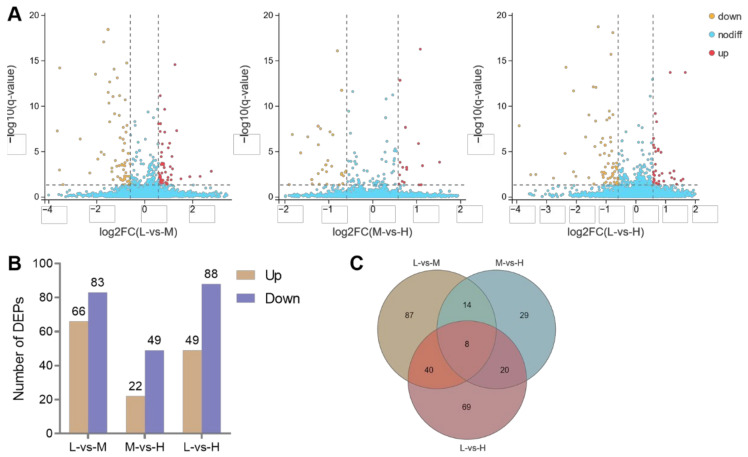
Differential expression proteins (DEPs) in three different altitudes of Tibetan sheep lungs. Volcano plots of DEPs (**A**), Number of DEPs (**B**), and Venn diagrams of DEPs (**C**) in three comparisons: L-vs-M, M-vs-H, and L-vs-H altitudes (*q* < 0.05 and fold change ≥ 1.5 or ≤ 0.66).

**Figure 9 animals-12-02134-f009:**
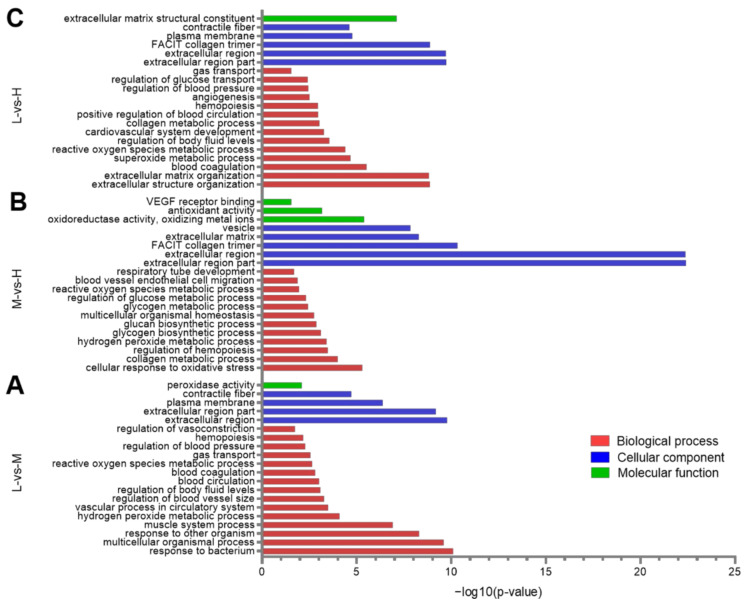
Histogram of Gene Ontology (GO) enrichment classification. A portion of the significantly enriched biological process (BP), cellular component (CC), and molecular function (MF) GO terms are shown, when comparing the DEPs between L-vs-M (**A**), M-vs-H (**B**), and L-vs-H altitudes (**C**).

**Figure 10 animals-12-02134-f010:**
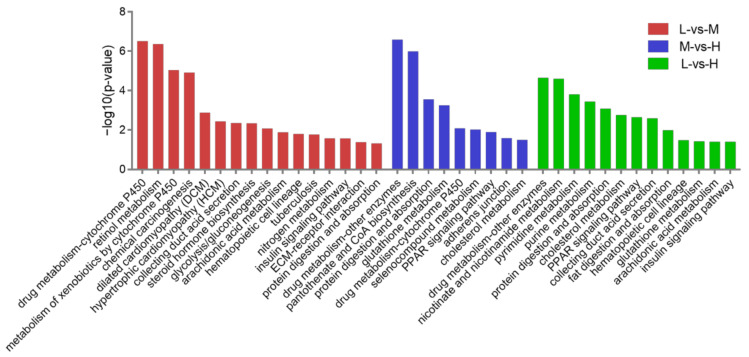
Kyoto Encyclopedia of Genes and Genomes (KEGG) pathway analysis. A portion of the significantly enriched pathways are shown, when comparing the DEPs between L-vs-M, M-vs-H, and L-vs-H.

**Figure 11 animals-12-02134-f011:**
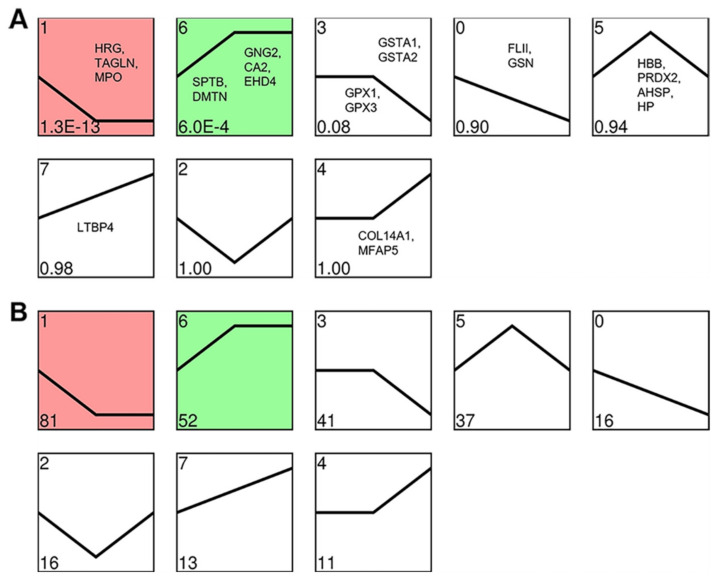
Cluster analysis of the expression patterns of DEPs in Tibetan sheep lungs with increasing altitude. Expression patterns are shown based on their *p*-value (**A**) and number (**B**) of DEPs in the lower left corner, colored patterns indicate significantly enriched patterns with altitude (*p* < 0.05). The proteins in the patterns are the key DEPs enriched in the corresponding patterns. Histidine rich glycoprotein (HRG), Transgelin (TAGLN), Myeloperoxidase (MPO), Spectrin beta chain, erythrocytic (SPTB), Dematin (DMTN), Guanine nucleotide binding protein subunit gamma 2 (GNG2), Carbonic anhydrase 2 (CA2), EH domain containing protein 4 (EHD4), Glutathione S-transferase A1 (GSTA1), Glutathione S-transferase A2 (GSTA2), Glutathione peroxidase 1 (GPX1), Glutathione peroxidase 3 (GPX3), Flightless 1 (FLII), Gelsolin (GSN), Hemoglobin subunit beta (HBB), Peroxiredoxin 2 (PRDX2), Alpha hemoglobin stabilizing protein (AHSP), Haptoglobin (HP), Latent transforming growth factor beta binding protein 4 (LTBP4), Collagen alpha-1(XIV) chain (COL14A1), Microfibrillar associated protein 5 (MFAP5).

**Figure 12 animals-12-02134-f012:**
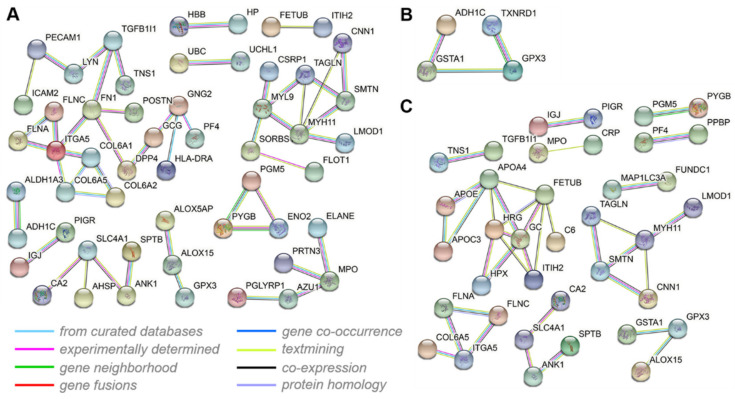
The Protein-protein interaction (PPI) networks of DEPs in three comparisons: L vs. M (**A**), M vs. H (**B**), and L vs. H (**C**). Nodes represent proteins, and lines of different colors represent the predicted different associations.

**Figure 13 animals-12-02134-f013:**
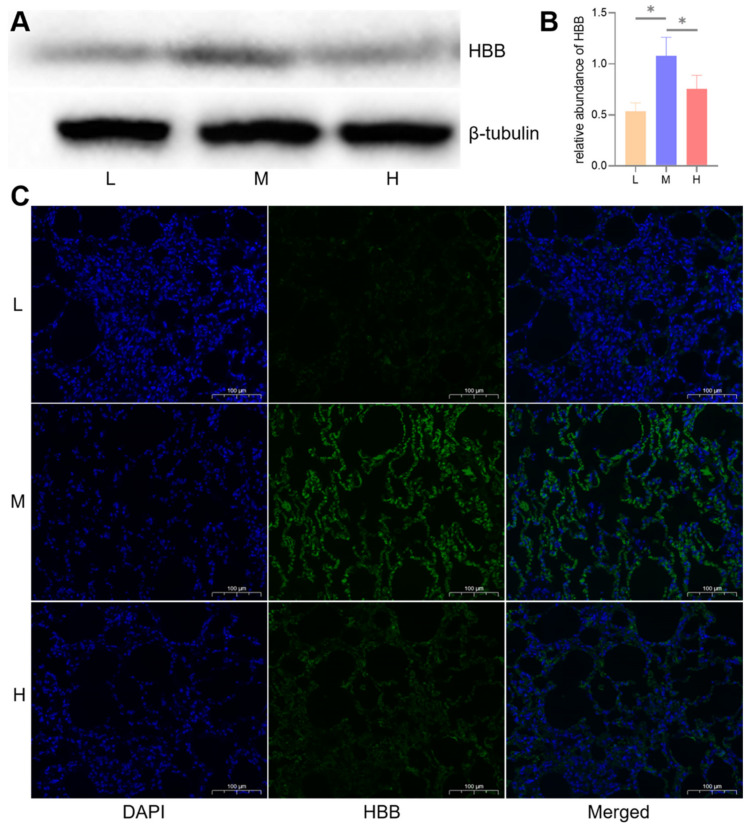
WB and immunofluorescence analysis of the key protein, HBB. Changes in the expression levels of HBB protein at low (L), middle (M), and high (H) altitudes, depicted by a WB (**A**), a histogram (**B**), and immunofluorescence (**C**), * indicates significant differences between different altitudes (*p* < 0.05).

## Data Availability

The mass spectrometry proteomics data have been deposited to the ProteomeXchange Consortium (accessed on 30 December 2021, http://proteomecentral.proteomexchange.org) via the iProX partner repository [58], with the dataset identifier PXD030661.

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
