# Peer review of "Physiology and Proteomic Basis of Lung Adaptation to High-Altitude Hypoxia in Tibetan Sheep"

_animals, 2022, doi:10.3390/ani12162134_

Round 1
Reviewer 1 Report
Dear Authors, The manuscript presented shows a lot of work and the data is definitely interesting. It appears challenging to create the narrative with relatively few (presumably outbred) sheep. Some tests provide lots of data points and some give very few (seemingly by design and low replication).
Some further tightening of the discussion is warranted, perhaps including reference to the qualitative / quantitative spread or spectrum of the data.

Author Response
Dear Reviewer, thank you for your review and suggestions.
You have pointed out very critical issues in this manuscript, like proteome sequencing and statistical analysis of lung physiology with smaller numbers of sheep. Although the sample sizes in this manuscript meet accepted statistical requirements, but indeed, the conclusions drawn would have been more reliable if the sample sizes were larger. These problems, due to lack of funding and difficulty in collecting samples, are indeed irreparable flaws in this manuscript.
In addition, you pointed out many grammatical errors and ambiguities, as well as many key problems in the text, and provided suggestions for improvement. This is of great help in improving the quality of this manuscript.
In response to these problems and suggestions, we have made substantial amendments to the manuscript during the revision period, including:
- Rewrote some sentences that were wrong or unclear, improved the language.
- Revised the manuscript according to the problems you pointed out, especially the discussion part, and replied in the attachment.
The revised manuscript and attachments have been uploaded for your review.
Thank you and best regards.
Pengfei Zhao

Reviewer 2 Report
This is a comprehensive study of adaptation of Tibetan Sheep to high altitude, with an examination of sheep from three different altitudes. The report shows a range of adaptations to high altitude, with non-linear responses to altitude in some functions.
The Introduction covers a wide range of previous work in this field for many species, noting that most studies only include two altitudes, assuming linear changes between the two levels.
The Methods covers a wide range of tests for the physiology of adaptation to low oxygen levels and an extensive study of proteomic differences.
The Results show that some changes seen at moderate altitudes, between 2500 to 3500 m are not extended to 4500 m, but the highest altitude requires different adaptations.
The Discussion puts these results in context with previous studies, showing consistency with other species where relevant.
Minor notes
The writing is good, but there are a few minor grammatical errors. The final version should be checked before publication. Some examples are given here.
Line 54. Replace “and” with “but”
Line 63. “Tibetan sheep are…”
Line 72. “proteins” not “protein”, “data” not “dates”.
Line 121-122. “One…tissue…was selected”. Or rephrase to remove “One” to keep “…tissues… were selected”.
Line 130-132. The sentence changes tense partway through. Rephrase to keep as a description of what happened, not an instruction of what to do.
Line 133. “One… specimen…was”, not “were”
Line 144. Change “dithiotreitol” to “dithiothreitol”.
Line 274. “approximately” not “approximate”, and “consisted of” not “were consisted by”.
Line 482. “data” not “date”
Author Response
Dear Reviewer, thank you for your review and suggestions.
You have pointed out many errors or ambiguities in the grammar, and provided suggestions for improvement, which have greatly helped to improve the quality of the text. Once again, thank you.
During the revision period, we made substantial amendments to the manuscript, included:
- rewrote some sentences that were too long or unclear.
- improved some grammatical errors.
The revised manuscript has been uploaded for your review.
Thank you and best regards.
Pengfei Zhao